# Mycotoxin Exposure during the First 1000 Days of Life and Its Impact on Children’s Health: A Clinical Overview

**DOI:** 10.3390/toxins14030189

**Published:** 2022-03-04

**Authors:** Paula Alvito, Luís Pereira-da-Silva

**Affiliations:** 1Food and Nutrition Department, National Institute of Health Dr. Ricardo Jorge, 1649-016 Lisbon, Portugal; 2Centre for Environmental and Marine Studies (CESAM), University of Aveiro, 3810-193 Aveiro, Portugal; 3Comprehensive Health Research Centre, NOVA Medical School|Faculdade de Ciências Médicas, 1169-056 Lisbon, Portugal; l.pereira.silva@nms.unl.pt; 4Hospital Dona Estefânia, Centro Hospitalar Universitário de Lisboa Central, 1169-045 Lisbon, Portugal

**Keywords:** early-life exposure, mycotoxins, children, adverse pregnancy outcomes, baby foods, breast milk, human biomonitoring

## Abstract

The first 1000 days of life are very sensitive to any event that alters health programming, and they represent a window for intervention to improve population health. Pregnant women, fetuses, and infants are particularly vulnerable to exposure to food contaminated with mycotoxins. This review aimed to gather data from the literature on mycotoxins exposure during intrauterine life and early childhood, and associated health risks, as assessed through human biomonitoring and mycotoxins occurrence in foods, in different continents. Maternal internal exposure to aflatoxins is associated with fetal growth restriction, while exposure to fumonisins increases the risk of offspring’s neural tube defects. Mycotoxin contamination of breast milk is reported worldwide, but data on adverse effects of the lactational transfer of mycotoxins on infant health are lacking. Young children are exposed to mycotoxins through contaminated infant formulas and baby foods. Both external and internal exposure to aflatoxins and fumonisins in children are reported to be associated with growth impairment. In low-income settings, where other co-factors can affect growth, this association should be interpreted with caution. Further studies on human biomonitoring of mother–infant pairs and young children are needed to guide management strategies aiming to minimize mycotoxin exposure at critical developmental stages.

## 1. Introduction

### 1.1. Mycotoxins Exposure in Early-Life

Mycotoxins are natural low-molecular-weight metabolites produced by fungal species that can be toxic for humans and animals. Food ingestion is the main route of exposure to this substance group, but inhalation and dermal contact could also contribute, although to a lesser extent. Mycotoxins are resistant to food processing and cooking practices and may be toxic even in low concentrations [1,2]. They occur all over the world and are closely associated with agricultural crops, particularly cereals, although they also appear in fruits, vegetables, and animal products (meat, dairy, eggs). They can affect single or multiple target organs, such as the liver, kidneys, and the immune system, with varying degrees of mutagenic, teratogenic, carcinogenic, and/or immunosuppressive potency [3,4]. Several mycotoxins may be carcinogenic to humans, as evaluated by the International Agency for Research on Cancer [5]. Under the climate change scenario, some fungal species might shift their geographical distribution in response to global warming, leading to changes in the pattern of mycotoxin occurrence and, thus, increasing human mycotoxin exposure [6,7].

Five major mycotoxins groups were reported to affect human health—aflatoxins (AFs) (mainly aflatoxin B_1_ (AFB_1_)), which cause liver cancer and have been implicated in child growth impairment and acute toxicoses; ochratoxins (OTAs) (mainly ochratoxin A), which are associated with renal diseases; fumonisins (FBs) (mainly fumonisin B_1_), which are associated with esophageal cancer and neural tube defects; trichothecenes (mainly desoxynivalenol (DON)), which are immunotoxic and cause gastroenteritis; and zearalenone (ZEN), which is known to have estrogenic effects [3,8]. *Aspergillus* mold strains produce AF and OTA, and *Fusarium* mold strains produce FBs, DON, and ZEN [9,10]. Primary food sources for AF include nuts, spices, corn (maize), cacao, coffee, rice, and cows’ milk. Cereals, coffee, and wine are a source of OTA, corn is a source of FB, wheat and barley are sources for DON, and wheat, barley, and sorghum are sources for ZEN [1,10]. Nevertheless, AFs and FBs are the most relevant mycotoxins that cause recognized harmful effects in fetuses and children [10,11].

Different countries throughout the world set regulations to protect consumers from food contaminated with mycotoxins. The Joint FAO/WHO Expert Committee on Food Additives reported the estimated mean dietary exposures to AFs, the most hepatotoxic substances known, in the ranges of 0.93–2.45, 3.5–180, and 0.3–53 ng/kg body weight/day for consumers in Europe, Africa, and in Asia, respectively. The contamination of food and feed by mycotoxins represents a serious health problem for humans and animals, as well as a considerable economic obstacle in most of the low- and middle-income countries [2].

Besides the free or parent mycotoxins (unchanged forms) that are regulated, several other non-regulated toxins are reported as modified and emerging mycotoxins, as well as a combination of mycotoxins that could impact human health [12,13,14], deserving special attention for their effects on pregnant women and young children. For all of the above reasons, understanding and monitoring human exposure to mycotoxins, particularly mothers and young children, is a key concern.

Children are much more vulnerable to the toxic effects of mycotoxins compared to adults because they have lower body mass and do not have a mature detoxification system [15,16]. The first 1000 days of life, the period from the fetal stage to 2 years old, is characterized by accelerated growth and developmental plasticity [17]. This early-life period is very sensitive to any event that alters the programming of the main body functions and represents a window for intervention to improve child and population health [18]. In this regard, pregnant women, fetuses, and infants are particularly vulnerable to exposure to environmental factors, particularly to food contaminants [18]. Mycotoxins from maternally-contaminated food can cross the placental barrier and affect fetal systems [19]. After birth, breastfed infants can be exposed to mycotoxins through contaminated breast milk [20]. In the first years of life, hazardous exposure can also occur when infants consume mycotoxin-contaminated infant formulas, fruit-based products, and, particularly, cereal-based baby foods [21,22].

An additional problem in assessing the effect of mycotoxins exposure effect on children’s health is the nonspecific nature of signs and symptoms attributable to the toxic properties of mycotoxins and the difficulty in considering them in the differential diagnosis of many conditions [23,24]. Mild forms of mycotoxicosis comprise rash, conjunctivitis, mouth ulcers, epistaxis, apnea, cough, wheezing, nausea, and vomiting, while severe forms include pulmonary hemorrhage, recurrent apnea, “pneumonia”, and bone marrow failure [23].

### 1.2. Human Biomonitoring and Health Risks

Assessing the exposure of pregnant women and children to mycotoxins can be carried out through human biomonitoring by direct measurements of biomarkers in biological samples (internal exposure) and determining the presence of mycotoxin levels in foods (external exposure) [25,26]. Assessing the presence of mycotoxins and concentration of their metabolites in human samples, such as urine, serum, plasma, and breast milk samples [27,28], aggregates exposure from different sources and by different exposure routes. Hence, human biomonitoring data provides a more accurate estimation of the body burden and can improve risk assessment [29]. In turn, age-stratified human biomonitoring data may be a useful tool for identifying environmental agents, such as mycotoxins, that may be of concern for vulnerable populations, such as children and pregnant women. Combined with additional efforts to identify potential sources of exposure, human biomonitoring could assist policymakers in prioritizing their actions to reduce mycotoxin exposure [30].

The analysis of biomarkers of exposure in biological matrices has become a common method in determining exposure to different mycotoxins [27,28,31]. Urine analysis presents some advantages because sampling is non-invasive and collection is easy; however, it reflects day-to-day variations in mycotoxin intake and, therefore, samples should be taken at different times over a 24 h period. Serum and plasma matrices have the advantage of requiring less sensitive methods because they contain higher levels of compounds. In addition, while urinary excretion normally indicates recent mycotoxin intake, plasma and serum measurements indicate long-term exposure. However, they are limited in that they require invasive collection methods and medical professionals. Breast milk can be used to monitor lactating women; nonetheless, it is an excellent source of information for exposure to breastfed infants [28]. In mothers and children, serum/plasma AF-albumin (AF-alb) and aflatoxin B_1_-lysine (AFB_1_-lys) adducts and breast milk AFM_1_ were also used as biomarkers for AF exposure [10]. There is a variety of selective and sensitive techniques for mycotoxin biomarker determination. High-performance liquid chromatography is widespread because of its superior performance and reliability compared to thin-layer chromatography with a high quality of separation and low limits of detection. Multiple detection systems may be coupled with chromatography, including fluorescence, ultra-violet, diode-array, electrochemical, mass spectrometry, and tandem mass spectrometry, which has advanced in the last years to the status of the reference in the field of mycotoxin analysis [27,32,33]. Biomarker analyses may, however, not be a perfect method for assessing mycotoxin exposure, as blood levels depend on some important factors, such as the narrow detection window for most toxins and their bioavailability in biologic fluids [34]. In a systematic review evaluating associations between dietary mycotoxin exposure and young children’s health, the reliability of methods for mycotoxin detection was a limitation to drawing conclusions [35].

## 2. Objective

The aim of this narrative review was to gather data from the literature on internal and external exposure to mycotoxins and their short- and long-term health risks during intrauterine life and early childhood for vulnerable populations, such as pregnant women and young children, in different continents.

## 3. Literature Search and Inclusion Criteria

An extensive search in the databases of medical literature, including Pubmed/Medline, Scopus, Science direct, Scielo, Medscape, the Cochrane Database of Systematic Reviews, and ClinicalTrials.gov, was performed over the last decade (2011–2021), and also included complementary information available from Google.

The key terms used for the literature search were “aflatoxin” OR “deoxynivalenol” OR “fumonisin” OR “mycotoxicosis” OR “mycotoxin” OR “ochratoxin” OR “patulin” OR “T-2 toxin” OR “zearalenone”. To refine the search, these were combined with (AND) the key terms “baby food contamination” OR “biomarkers” OR “biomonitoring” OR “breast milk” OR “breastfeeding contamination” OR “cereal-based” OR “early-life exposure” OR “fetal exposure” OR “fruit-based” OR “lactating mothers” OR “pregnant women” OR “public health”.

Two review authors (PA and LPdS) independently screened all the titles and abstracts identified by the search strategy. Special attention was given to studies addressing adverse effects or potential adverse effects of mycotoxin exposure on fetal health and the health of children up to 24 months of age. The critical assessment for each study followed proposed guidelines for narrative style literature reviews [36] and included the key results, limitations, suitability of the methods to test the hypothesis, quality and interpretation of the results, and impact of the conclusions on the human health.

This review included 35 main studies and documents (Appendix A) assessing the exposure to mycotoxins of pregnant women and children up to two years of age, by the presence of mycotoxin biomarkers of exposure in biological matrices and mycotoxin levels in foods.

## 4. Prenatal Exposure to Mycotoxins

A systematic review of 17 epidemiological studies evaluated the relationship between adverse pregnancy outcomes and maternal mycotoxin exposure [11]. All studies were observational; ten were conducted in the sub-Saharan African region, three in Europe, two in North America (the United States and Mexico), and two in the United Arab Emirates. In most of the studies, exposure to mycotoxins was determined by human biomonitoring, and in remaining indirect approaches were used based on information regarding agricultural/weather conditions or a validated aflatoxigenic food frequency questionnaire. Regarding offspring outcomes, the majority of studies were focused on aflatoxigenic food consumption. From this systematic review [11] eleven studies (nine from sub-Saharan Africa and two from the United Arab Emirates) reported adverse pregnancy outcomes associated with mycotoxin quantitatively measured in maternal blood, cord blood, and infant blood samples, with values ranging between 0.44 and 285,000 pg/mL, 4 and 238,177 pg/mL, and 17 and 165,067 pg/mL, respectively, as shown in Table 1. The aflatoxin group (range of 0.44–238,177 pg/mL) was the one with more studies performed (9/11), followed by AFT and OTA (200–3500 pg/mL) and FB_1_ (0.45–2.85 µg/mL), with one study each. Most studies used HPLC methods for mycotoxin biomarker quantification. Low birth weight is the most widely adverse pregnancy outcome reported (8/11), followed by neonatal jaundice (4/11), and pre-eclampsia and preterm birth (1/11).

Good evidence was found on an adverse effect of maternal AF exposure on fetal growth, indicated by a decreased mean birth weight and an increased risk of low birth weight among exposed newborn infants [11]. Two studies [37,38] reported this effect only in female neonates. In one study, it was found that intrauterine exposure to AFs was associated with an adverse effect on growth during the first year of life [39]. Nevertheless, in several studies included, analyses were not sufficiently adjusted for other risk factors for intrauterine growth restriction, such as maternal infections and socioeconomic status, which may be associated with AF exposure in pregnancy and can lead to confounding results [11]. Plausible explanations for the impairment of fetal growth related to maternal AF exposure include direct and indirect toxicity via maternal systemic inflammation, impaired placental growth, and/or elevation of placental cytokines [9]. One study reported the presence of OTAs in cord blood samples from pregnant women, in combination with AFs [38]. It was suggested that OTA is more likely to be detected in the cord blood of girls than boys and that, when it is present in combination with AFs and their metabolites (aflatoxicol), the birth weight is likely to be depressed; however, it was not possible to reach definite conclusions due to the small sample size [38].

Maternal AF exposure during pregnancy may also increase the likelihood of neonatal jaundice. In two studies [40,41], maternal AF exposure during pregnancy increased the likelihood of neonatal jaundice by more than two-fold in jaundiced neonates compared to non-jaundiced neonates, while this association was not found in two other studies [42,43].

An etiological role in preeclampsia was associated with FB exposure among maize-consuming pregnant women and with neural tube defects in the offspring [44,45]. In one of the studies [44], maternal exposure to FBs was assessed based not only on the reported number of corn tortillas consumed during the first trimester of pregnancy but also on the measured FBs from maternal blood. In animal models, it was demonstrated that FBs disrupt the biosynthesis of sphingolipids, which interferes with folate receptors, affect folate bioavailability, and ultimately causes neural tube defects [46,47].

**Table 1 toxins-14-00189-t001:** Adverse pregnancy outcomes and maternal-fetal mycotoxin biomarkers in blood, in different continents [11].

Adverse Pregnancy Outcomes	Continent	Country	N	Mycotoxin	Mycotoxin Concentration Range (pg/mL or pg/mg Albumin)			Analytical Method	References
					Maternal Blood	Cord Blood	Infant Blood		
Birth weight/low birth weight	Africa	Kenya	208	AF	12–11,574	17–6819	-	HPLC	[37]
		Sierra Leone	64	AF	-	4.0–9000	-	HPLC	[38]
				OTA		200–3500	-		
		Nigeria	164	AFB_1_	-	32,300–35,600	-	TLC	[40]
		Gambia	138	AF	4.8–260	5.0–89.6	-	ELISA (AF-alb)	[39]
		Ghana	785	AFB_1_	0.44–268.73	-	-	HPLC (AFB_1_-lys)	[46]
		Uganda	220	AFB_1_	0.71–95.60	-	-	HPLC (AFB_1_-lys)	[47]
	Asia	UAE	201	AF	-	110–15,225	-	HPLC	[48]
							-		
		UAE	166	AFM_1_	30–8490	50–10,440	-	HPLC	[42]
Neonatal jaundice	Africa	Nigeria	407	AF	17–165,067	-	17–165,067	HPLC	[41]
			202	AF	-	13–238,177	24–11,728	HPLC	[43]
			164	AFB_1_	-	32,300–35,600	-	TLC	[40]
	Asia	UAE	166	AFM_1_	30–8490	50–10,440	-	HPLC	[42]
Pre-eclampsia/eclampsia	Africa	South Africa	51	FB_1_	450,000–2,850,000			HPLC	[44]
Preterm birth		Ghana	785	AFB_1_	0.44–268.73			HPLC (AFB_1_-lys)	[46]

AF—aflatoxins group, including aflatoxin B_1_, B_2_, G_1_, G_2_, M_1_, M_2_, and aflatoxicol (AFl); AF-alb—aflatoxin albumin adduct; AFB_1_-lys—aflatoxin B_1_-lysine adduct; HPLC—high-performance liquid chromatography, TLC—thin layer chromatography, ELISA—enzyme-linked immunosorbent assay.

Three of the studies [46,47,48] found a higher likelihood of preterm birth among pregnant women exposed to AFs, although other studies did not confirm this association.

## 5. Infant Exposure through the Mother’s Milk

Lactational transfer of mycotoxins occurs via the maternal diet in unaltered or metabolized forms into breast milk, depending on the mother’s eating habits [22].

Peraica et al. [16] and recently, Hernandez et al. [22] and Coppa et al. [49], reviewed in detail the presence of AFM_1_ and OTA and its concentration in breast milk from several continents. As shown in Table 2, the most studied and frequently detected mycotoxins in breast milk samples were AFM_1_ and OTAs [16,22,49]. According to these reviews, the highest AFM_1_ values in breast milk were reported in Egypt [50] followed by Asia (United Arab Emirates) [48], Europe (Serbia) [51], and America (Ecuador) [52]. All these values largely exceeded the maximum limit of AFM_1_ in raw milk adopted by European countries (50 ng/L) [53,54], hence confirming a high rate of exposure of newborns to AFM_1_. Concerning OTAs, the highest concentrations were found in breast milk samples from Africa (Serra Leone) [38], followed by Europe (Turkey) [55] followed by Asia (Iran) [56], and America (Chile) [57]. AFM_1_ in breast milk is frequently assessed by enzyme-linked immunosorbent assay (ELISA) kits [50,58] that were developed and validated for animal milk and are directly applied to human milk, which is a very different matrix, with no reported LOD or limit of quantification (LOQ) values.

A recent comprehensive review on AFM_1_ exposure in Africa concluded that 85%–100% of African infants were highly exposed to AFM_1_ through human milk intake, specifically in Egypt, Kenya, Morocco, Nigeria, and Tanzania [59] (Table 2). Other studies also confirm the presence of this biomarker of exposure in America, such as Mexico [60] and Brazil [61].

A recent study in Nigeria assessed mycotoxin co-exposure in 23 mother–infant pairs, analyzing plate-ready food, breast milk, and urine samples of mothers and their exclusively breastfed infants [62]. According to the authors, this was the first study to examine mycotoxins in breast milk in combination with maternal food intake and biomarker analysis in urine. This study revealed the co-occurrence of two to five mycotoxins including regulated (AFs, DON, and OTA) and emerging mycotoxins (alternariol monomethyl ether and beauvericin, dihydrocitrinone, and enniatin B) [62]. The mycotoxins detected in breast milk included AFM1, OTA, and emerging mycotoxins. A moderate association of OTA was found in milk and in the urine of mothers and infants but no other significant correlations were found. According to the authors, this is the first comprehensive assessment of mycotoxin-mixtures across different food/biological matrices in mother–infant pairs and demonstrated that mycotoxin co-occurrence in food, breast milk and urine is frequent [62]. The presence of other mycotoxins as zearalenone and deoxinivalenol, together with AFM1 and OTA, were recently detected in breast milk samples from Turkey [63] and zearalenone from Iran [64].

The most frequently used techniques for AFM_1_ detection in breast milk with sample preparation are mainly based on liquid chromatography coupled with a fluorescence detector (HPLC/FD) and, more recently, coupled with mass spectrometry (LC-MS/MS) and high-resolution mass spectrometry (LC-HRMS) approaches [32].

Consumed foods more likely to be contaminated with AFM_1_ are cereals, spices, seeds, nuts, and dairy products [22]. A study from Norway indicated that OTA levels in breast milk were strongly correlated with the consumption of liver paste and cake [22]. In central Mexico, 89% of breast milk samples from nursing mothers contained AFM_1_ which was significantly associated with consumption of egg, cola drinks, and sunflower oil [60]. It can be postulated that the geographical location is a relevant factor influencing the eating habits, culinary culture, and diet of lactating women and, consequently, the mycotoxin contamination of breast milk [49].

A significant difficulty is determining the average daily AF intake of breastfed infants through breast milk because the maternal intake of AFs varies from day to day, and breast milk levels fluctuate [20]. In addition, the specificity and sensitivity of analytical methodologies used to detect mycotoxins in human milk were questioned [32].

Although reasonable knowledge has been acquired worldwide on mycotoxin contamination of breast milk, no study to date has assessed the adverse effects of the lactational transfer of mycotoxins on infant health. The clarification of this issue is important since breastmilk is the gold standard, particularly for infants living in low- and mid-income countries. Assessment of the independent effects of contaminated breast milk on breastfed infants’ health is difficult due to the coexistence of several relevant factors affecting the infants’ health. Meanwhile, a reduction in mycotoxin contamination of lactating mothers and the consequent reduction in infant exposure through breast milk is desirable [20].

## 6. Other Sources of Mycotoxin Exposure during Early Childhood

The two aforementioned reviews [22,49] also evaluated the mycotoxin exposure of infants from birth to 9 months of age through the consumption of infant formulas, cereal, and fruit-based products. The level of infant exposure by the estimated daily intake of mycotoxins was determined from the consumption of foods. To estimate risk, the estimated daily intake was compared with the tolerable daily intake established by EFSA [22]. The European Commission has established maximum levels for mycotoxins, namely for AFB_1_, OTA, DON, ZEN, and the sum of FB_1_ and FB_2_, in processed cereal-based foods and baby foods for infants and young children at 0.100 µg/kg, 0.50 µg/kg, 200 µg/kg, 20 µg/kg, and 200 µg/kg, respectively. For AFM_1_, the limits established for infant formulas and follow-on formulas is 0.025 µg/kg, and for raw milk, heat-treated milk, and dairy products, the limit is 0.050 µg/kg [53,54].

### 6.1. Mycotoxin Contamination of Infant Formulas

The prevalence of mycotoxins in infant formulas is different among different countries [22,49,61]. In Europe, AFM_1_, AFB_1_, OTA, and ZEN were the main mycotoxins detected in Italy, Portugal, Spain, and Turkey [15,65,66,67,68] exceeding the limits established by the European Union [55,56]. Contamination of infant formulas was also reported in American continents (Brazil and Canada) with AFM_1_, AFB_1_, and OTAs, in South Korea with ZEN, and in Burkina Faso with AFB_1_, OTAs, and FBs [22,49,62], and with AFM_1_ in Egypt [59].

### 6.2. Mycotoxin Contamination of Cereal-Based Products for Infants

In European countries (Italy, Portugal, Spain, and Turkey), the contamination of cereal products for infants with mycotoxins regulated in the European Union, such as aflatoxins, OTAs, patulin (PAT), and *Fusarium* toxins, was reported [15,22,49]. These cereal products were also contaminated with non-regulated mycotoxins, such as nivalenol, beauvericin, sterigmatocystin, eniantins, and fumonisin metabolites [22,49].

In American countries (Canada and the United States), the detection of OTAs, ZEN, AFB_1_, AFB_2_, and AFG_2_, with some levels exceeding the European Union maximum limits, was reported [22,49]. In Asia, contamination with AFB_1_ was reported in Iran and with OTAs in Syria [22].

In African countries, infants’ exposure to AFM_1_ through consumption of milk and milk derivatives is a matter of concern, with samples analyzed exceeding the European regulations in Zimbabwe, Nigeria, Sudan, and Egypt [59].

In some countries, the detected levels of mycotoxins did not exceed the maximum limits set in Europe for cereal-based foods for infants [22,69].

### 6.3. Mycotoxin Contamination of Fruit-Based Products for Infants

The incidence of mycotoxins in baby fruit purees, compote, and juices is not very high, and patulin (PAT) was the only mycotoxin detected among several countries, except for OTAs in Syria [22,49]. Important potential factors influencing this contamination are the processing stages and the type of storage carried out in apple-based products since the processing and washing stages of raw materials reduce the PAT content in apple juice [70].

In Nigeria, another study assessing risks from consumption of complementary foods by young children [71] found that aflatoxins and fumonisins co-contaminated 42% of the cereal- and nut-based food samples at mean concentrations exceeding the European Union limits [53,54].

## 7. Adverse Health Effects from Early Childhood Mycotoxin Exposure

In some African countries, growth impairment has been associated with mycotoxin exposure in early childhood. In Tanzania, a cross-sectional study evaluated growth in 215 infants exposed to FB_1_, FB_2_, and FB_3_ in maize flour during the first 6 months of age [72]. At 12 months of age, infants exposed to FB intakes above 2 mg/kg were significantly shorter by 1.3 cm and 328 g lighter [72]. Additionally in Tanzania, a longitudinal study on 166 infants investigated the association between growth and exposure to AFs and FBs of infants between 6 and 14 months of age [73]. Urinary FB_1_ concentrations at recruitment were significantly associated with lower length-for-age z-scores both at 6 months and 12 months after recruitment. In each, the mean urinary FB_1_ at recruitment and at 6 and 12 months from recruitment was negatively associated with lower length-for-age z-scores and length velocity at 12 months after recruitment. The negative association between plasma AF-alb and growth did not reach statistical significance [73]. In Kenya, a cross-sectional study on 242 infants found an association between the consumption of aflatoxin-contaminated flour by infants between 6 and 36 months of age and growth impairment [74]. Being underweight was more frequent among AF consumers than AF non-consumers (41.4% vs. 27.3%, *p* < 0.05), as it was for stunting (32.4% vs. 28.9%, *p* < 0.05) [74]. In Benin and Togo, a cross-sectional study on 480 infants evaluated the association between exposure to aflatoxins of infants between 9 months and 5 years of age and growth [75]. Significant negative correlations between individual AF-alb concentrations and height-for-age, weight-for-age, and weight-for-height were found. In a categorical analysis, the association with AF-alb concentrations was significant with dose–response relations with height-for-age and weight-for-age z-scores [75]. A recent systematic review assessed dietary mycotoxin exposure and child growth, immune system, morbidity, and mortality [23]. Overall, the certainty of the estimates for the association between dietary mycotoxin exposure and child health was very low, mainly because of the risk of bias and inconsistency. This might be due to considerable heterogeneity among studies, variation in measurement methods, exposure period, differences in study populations, limited sample sizes, and failure to adjust for confounders. Regarding the latter factor, data from low- and middle-income countries should be interpreted with caution since, in these settings, important co-factors other than mycotoxin exposure can affect infant growth, such as socioeconomic status, limited food access, and endemic infections [76]. In addition, the mechanisms underlying impaired growth due to AF exposure are still unclear. Intestinal function damage, reduced immune function, and alteration in the insulin-like growth factor axis caused by liver damage are the suggested hypotheses [77]. As both AFs and child undernutrition is common in sub-Saharan Africa, the interactive relationship between both factors needs to be better understood for an effective intervention aimed at reducing undernutrition [77].

Since the first evidence of the possible role of mycotoxins, particularly OTAs, was discussed in the pathobiology of autism in Italian children [78], the scientific community has increased its interest in this research domain. Recently, in Spain, the levels of 19 mycotoxins in plasma samples from healthy and sick children (digestive, autism spectrum, and attention deficit hyperactivity disorders) were determined [16]. OTA was the most prevalent mycotoxin in all groups and was higher in healthy children than sick children. Sterigmatocystin was detected in all groups, especially in sick children [16].

## 8. Conclusions

The first 1000 days of life are very sensitive to any event that alters the health programming of the main body functions, and they represent a window for intervention to improve child and population health. Pregnant women, fetuses, and infants are particularly vulnerable to exposure to environmental factors, particularly food contaminated with mycotoxins.

Human biomonitoring is considered a useful tool that can be used as a third-tier method for the refinement of exposure assessment. Data on mycotoxin biomarkers collected during intrauterine life and early childhood in several countries and continents illustrate well the importance of using this tool to identify potential associations between adverse health risks and maternal exposure to mycotoxins, namely aflatoxins, OTA and FB_1_. There is good evidence that maternal dietary aflatoxin exposure is associated with fetal growth impairment, while prenatal exposure to fumonisins may be associated with neural tube defects in offspring.

Mycotoxin levels in breast milk are good predictors of breastfed infants’ exposure. However, the relation between maternal intake and breast milk transfer is poorly understood. In addition, the risks of the lactational transfer of mycotoxins on infant health have not been demonstrated. Children’s exposure to hazardous mycotoxins also occurs when they consume contaminated infant formulas and baby foods, especially cereal-based baby foods. In low-income countries, the association between mycotoxin-contaminated food and growth impairment was reported but in these settings, effects of mycotoxin exposure on growth should be interpreted with caution since other co-factors can affect infant growth, such as poor socioeconomic status, limited food assessment, and endemic infection.

In conclusion, faced with generalized exposure to mycotoxins, efforts should be made by clinicians, health professionals, and the scientific community to spread and enhance the knowledge on mycotoxin exposure and associated health risks, particularly in early-life stages. Development of epidemiological studies gathering mothers and young children and human biomonitoring national programs for these vulnerable populations are needed to support risk management strategies to reduce mycotoxin exposure.

## Figures and Tables

**Table 2 toxins-14-00189-t002:** AFM_1_ and OTA highest concentration values (ng/L) in breast milk, in different continents [16,22,49].

Mycotoxin	Continent	Country	N	Concentration Range (ng/L)	Analytical Method	References
AFM_1_	Africa	Egypt	150	200–19,000	ELISA	[50]
	America	Ecuador	78	17–458	HPLC/FD	[52]
	Asia	United Arab Emirates	201	110–4060	HPLC/FD	[48]
	Europe	Serbia	60	5–570	ELISA	[51]
OTA	Africa	Serra Leone	113	200–337,000	HPLC/FD	[38]
	America	Chile	50	10–186	LC-MS	[57]
	Asia	Iran	84,171	110–7340	HPLC/FD	[56]
	Europe	Turkey	75	621–13,111	HPLC/FD	[55]

AFM_1_—aflatoxin M_1_, OTA—ochratoxin A, HPLC/FD—liquid chromatography with fluorescence detection, ELISA—enzyme-linked immunosorbent assay; LC-MS—liquid chromatography coupled with mass spectrometry.

## Data Availability

Not applicable.

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
