# Peer review of "Mycotoxin Exposure during the First 1000 Days of Life and Its Impact on Children’s Health: A Clinical Overview"

_toxins, 2022, doi:10.3390/toxins14030189_

Round 1
Reviewer 1 Report
The manuscript entitled “Mycotoxin exposure during the 1000 first days of life and its impact on child health: a clinical overview” to review on the effects of mycotoxin exposure during intrauterine life and early childhood. Overall, this review article is well written, the authors gather the possible significant information for providing a conclusion. In my opinion, this manuscript can be acceptable for publication.
Author Response
The manuscript entitled “Mycotoxin exposure during the 1000 first days of life and its impact on child health: a clinical overview” to review on the effects of mycotoxin exposure during intrauterine life and early childhood. Overall, this review article is well written, the authors gather the possible significant information for providing a conclusion. In my opinion, this manuscript can be acceptable for publication.
Response: Thank you very much for the positive comments.

Reviewer 2 Report
The authors have undertaken to describe an intriguing and important syndromic effect of childhood (including prenatal) exposure to mycotoxins. The preparation of a review paper on this topic is essential in the absence of extensive research on this topic.
The paper is based on publications found by the authors using recognised online publication databases. However, the list of keywords searched by the authors is too narrow.
There is a definite lack of such keywords as: "mycotoxins" "deoxynivalenol" "zearalenone" "T-2 toxin" "ochratoxin A" "biomarkers" Perhaps, for this reason, the publication lacked a more significant number of cited sources, e.g:
Braun D, Schernhammer E, Marko D, Warth B (2020) Longitudinal assessment of mycotoxin co-exposures in exclusively breastfed infants
Cantú-Cornelio F, Aguilar-Toalá JE, de León-Rodríguez CI, et al (2016) Occurrence and factors associated with the presence of aflatoxin M1 in breast milk samples of nursing mothers in central Mexico. Food Control 62:16–22 . doi: 10.1016/j.foodcont.2015.10.004
Ishikawa AT, Takabayashi-Yamashita CR, Ono EYS, et al (2016) Exposure assessment of infants to aflatoxin M1through consumption of breast milk and infant powdered milk in Brazil. Toxins (Basel) 8:1–11 . doi: 10.3390/toxins8090246
Ojuri OT, Ezekiel CN, Sulyok M, et al (2018) Assessing the mycotoxicological risk from consumption of complementary foods by infants and young children in Nigeria. Food Chem Toxicol 121:37–50 . doi: 10.1016/j.fct.2018.08.025
Considering these shortcomings, I encourage the authors to review the scientific literature better and complete this promising publication.
Author Response
Reviewer 2
The authors have undertaken to describe an intriguing and important syndromic effect of childhood (including prenatal) exposure to mycotoxins. The preparation of a review paper on this topic is essential in the absence of extensive research on this topic.
The paper is based on publications found by the authors using recognised online publication databases. However, the list of keywords searched by the authors is too narrow.
There is a definite lack of such keywords as: "mycotoxins" "deoxynivalenol" "zearalenone" "T-2 toxin" "ochratoxin A" "biomarkers" Perhaps, for this reason, the publication lacked a more significant number of cited sources, e.g:
Braun D, Schernhammer E, Marko D, Warth B (2020) Longitudinal assessment of mycotoxin co-exposures in exclusively breastfed infants
Cantú-Cornelio F, Aguilar-Toalá JE, de León-Rodríguez CI, et al (2016) Occurrence and factors associated with the presence of aflatoxin M1 in breast milk samples of nursing mothers in central Mexico. Food Control 62:16–22. doi: 10.1016/j.foodcont.2015.10.004
Ishikawa AT, Takabayashi-Yamashita CR, Ono EYS, et al (2016) Exposure assessment of infants to aflatoxin M1through consumption of breast milk and infant powdered milk in Brazil. Toxins (Basel) 8:1–11. doi: 10.3390/toxins8090246
Ojuri OT, Ezekiel CN, Sulyok M, et al (2018) Assessing the mycotoxicological risk from consumption of complementary foods by infants and young children in Nigeria. Food Chem Toxicol 121:37–50. doi: 10.1016/j.fct.2018.08.025
Considering these shortcomings, I encourage the authors to review the scientific literature better and complete this promising publication.
Response: Thank you very much for the insightful questions and suggestion. Accordingly, the authors have made a more complete literature search adding more specific key terms (Section 3. Literature Search and Inclusion Criteria). A new literature search resulted in selection of eight further high-quality studies [references 61, 67, 65, 66] for the review, which included the four suggested articles [references 62, 63, 64, 73].
Key studies and documents supporting the review are now presented in supplementary tables.
Reviewer 3 Report
This review describes the sources of exposure to mycotoxins, as well as (in a shorter manner) the adverse health effects from mycotoxin exposure. The manuscript is written in a coherent manner.
One issue to be resolved:
- Section 3 - Literature search: nothing is mentioned in regard to the criteria for inclusion of the articles. Although this article is not intended to be a meta-analysis, but a narrative review, minimum quality inclusion criteria should have been considered. I would have checked, for example, the correlation between the results / the description from a review and the conclusions, the coherence of the text etc. This is necessary because information available from Google was included and there are still minor journals (usually not indexed in the major databases) that have puplished over time many or fewer "lower quality" articles.
Author Response
This review describes the sources of exposure to mycotoxins, as well as (in a shorter manner) the adverse health effects from mycotoxin exposure. The manuscript is written in a coherent manner.
One issue to be resolved:
- Section 3 - Literature search: nothing is mentioned in regard to the criteria for inclusion of the articles. Although this article is not intended to be a meta-analysis, but a narrative review, minimum quality inclusion criteria should have been considered. I would have checked, for example, the correlation between the results / the description from a review and the conclusions, the coherence of the text etc. This is necessary because information available from Google was included and there are still minor journals (usually not indexed in the major databases) that have published over time many or fewer "lower quality" articles.
Response: Thank you very much for the insightful critic and suggestion. Accordingly, the Section 3 was renamed as Literature Search and Inclusion Criteria, in order to clarify details on the search strategy and critical assessment of the studies according to guidelines for narrative style literature reviews (reference 33). Coherence of objectives, results and conclusions was assessed in each study. Google rarely consulted, only for complementary information on articles and documents obtained from recognized scientific databases. In supplementary tables (S1-S3), the main articles used for the review are now presented with the respective references of high-quality peer review journals.
Round 2
Reviewer 2 Report
I accept the manusctipt in the present form.